# Using MALDI-TOF MS to Identify Mosquitoes from Senegal and the Origin of Their Blood Meals

**DOI:** 10.3390/insects14100785

**Published:** 2023-09-26

**Authors:** Fatou Kiné Fall, Adama Zan Diarra, Charles Bouganali, Cheikh Sokhna, Philippe Parola

**Affiliations:** 1Aix Marseille Univ, IRD, AP-HM, SSA, VITROME, 13005 Marseille, France; fakifa22@hotmail.fr (F.K.F.); adamazandiarra@gmail.com (A.Z.D.); 2IHU Méditerranée Infection, VITROME, 19–21 Boulevard Jean Moulin, 13005 Marseille, France; cheikh.sokhna@ird.fr; 3VITROME Dakar, Campus International IRD-UCAD Hann, Dakar 1386, Senegal; charles.bouganali@ird.fr

**Keywords:** MALDI-TOF MS, mosquitoes, *Anopheles*, *Culex*, *Aedes*, Senegal

## Abstract

**Simple Summary:**

Mosquitoes are capable of transmitting numerous diseases to humans and animals. However, not all mosquitoes are vectors. It is therefore very important to identify mosquitoes in order to distinguish vectors from non-vectors. Our aim was to evaluate the ability of MALDI-TOF MS to identify mosquitoes collected in Senegal and stored for several months in silica gel, and to determine the origin of their blood meal. We have identified 12 different species, most of which are vectors of human disease. We also found that these mosquitoes fed mainly on humans. Our results confirm that MALDI-TOF MS is a promising and rapid technique for identifying mosquitoes and the origin of their blood meal. However, the percentage of good-quality spectra was low, demonstrating that mosquitoes should be pre-treated before being preserved in silica gel. Transferring the MALDI-TOF MS database to Senegal will enable mosquitoes to be identified on the spot as quickly as possible.

**Abstract:**

Mosquitoes are arthropods that represent a real public health problem in Africa. Morphology and molecular biology techniques are usually used to identify different mosquito species. In recent years, an innovative tool, matrix-assisted desorption/ionization time-of-flight mass spectrometry (MALDI-TOF MS), has been used to identify many arthropods quickly and at low cost, where equipment is available. We evaluated the ability of MALDI-TOF MS to identify mosquitoes collected in Senegal and stored for several months in silica gel, and to determine the origin of their blood meal. A total of 582 mosquitoes were collected and analysed. We obtained 329/582 (56.52%) MALDI-TOF MS good-quality spectra from mosquito legs and 123/157 (78.34%) good-quality spectra from engorged abdomens. We updated our home-made MALDI-TOF MS arthropod spectra database by adding 23 spectra of five mosquito species from Senegal that had been identified morphologically and molecularly. These included legs from *Anopheles gambiae*, *Anopheles arabiensis, Anopheles* cf. *rivulorum*, *Culex nebulosus*, *Anopheles funestus*, and three spectra from abdomens engorged with human blood. Having updated the database, all mosquitoes tested by MALDI-TOF MS were identified with scores greater than or equal to 1.7 as *An. gambiae* (*n* = 64), *Anopheles coluzzii* (*n* = 12), *An. arabiensis* (*n* = 1), *An. funestus* (*n* = 7), *An. cf rivulorum* (*n* = 1), *Lutzia tigripes* (*n* = 3), *Cx. nebulosus* (*n* = 211), *Culex quinquefasciatus* (*n* = 2), *Culex duttoni* (*n* = 1), *Culex perfescus* (*n* = 1), *Culex tritaeniorhynchus* (*n* = 1), and *Aedes aegypti* (*n* = 2). Blood meal identification by MALDI-TOF MS revealed that mosquitoes had fed on the blood of humans (*n* = 97), cows (*n* = 6), dogs (*n* = 2), goats (*n* = 1), sheep (*n* = 1), and bats (*n* = 1). Mixed meals were also detected. These results confirm that MALDI-TOF MS is a promising technique for identifying mosquitoes and the origin of their blood meal.

## 1. Introduction

Mosquitoes are arthropods that represent a real public health problem due to the fact that they are capable of transmitting infectious agents. These infectious agents cause various pathologies in humans, including malaria, dengue, chikungunya, yellow fever, Japanese encephalitis, lymphatic filariasis, zika, and West Nile Virus [1,2]. The main mosquito vectors belong to three genera: *Anopheles*, *Culex,* and *Aedes* [3]. According to the World Health Organization (WHO), in 2020 malaria caused 627,000 deaths around the world [4]. Malaria is transmitted to humans through the bite of an *Anopheles* female infected with *Plasmodium* spp. [2]. Dengue is caused by one of the four serotypes of the Dengue virus and is transmitted to humans through the bites of mosquitoes of the genus *Aedes*, including *Aedes aegypti* and *Aedes albopictus*. It is responsible for some 40,000 deaths annually [2,5,6].

In Senegal, a total of 143 species of mosquitoes divided into 12 genera have been recently recorded [7]. The genera *Aedes*, *Anopheles*, and *Culex* are predominantly represented [7]. In 2019, the number of malaria cases in Senegal was estimated at 354,708 [8]. In 2018, Senegal experienced its largest Dengue epidemic with 342 cases being recorded [2]. These figures justify the importance of monitoring and fighting the mosquito vectors.

The methods currently used to identify mosquitoes include morphology and molecular biology [9]. Morphological identification is based on the observation of taxonomic criteria [10,11] but has limitations, including the lack of appropriate documentation such as dichotomous identification keys, the need for entomological expertise, and the difficulty of differentiating between species of the same complex [9,12]. Molecular biology has made it possible to identify many arthropods as well as the origin of their blood meals [13,14,15]. However, this method is limited by its high cost, the lack of universal primers to identify arthropods with certainty, and the incompleteness of the National Centre for Biotechnology Information (NCBI) GenBank database [9]. To determine the origin of blood meals in arthropods, serological techniques such as enzyme-linked immunosorbent assays (ELISA) and precipitation tests have also been used [16,17]. However, serological techniques have limitations, including the availability of specific antisera against a wide variety of host species, handling time, high cost, and antibody cross-reactions for closely related species [18].

After revolutionising clinical microbiology with the identification of bacteria, parasites, fungi, and archaea [19], MALDI-TOF MS has been recently proposed in medical entomology as an alternative tool for the rapid and cost-effective identification of many arthropods including mosquitoes, as well as the origin of their blood meal [9,20,21,22,23]. The aim of our study was to evaluate the ability of MALDI-TOF MS to identify mosquitoes collected in Senegal and stored for several months in silica gel and to determine the origin of their blood meals.

## 2. Materials and Methods

### 2.1. Mosquito Collection

All mosquitoes were collected in the Kédougou region of Senegal during one week (seven day) in September 2019. Specifically, collections were made in seven villages: Kédougou (12°55′ N, 12°18′ W), Bandafassi (12°33′ N, 12°17′ W), Boundoucoundi (12°31′ N, 12°20′ W), Indar (12°31′ N, 12°18′ W), Silling (12°32′ N, 12°16′ W), Ngari (12°38′ N, 12°15′ W), and Dambakoye (12°30′ N, 12°24′ W) (Figure 1). Two methods were used to collect mosquitoes: BG-Sentinel traps (Biogents AG, Regensburg, Germany) [24] and Residual Morning Fauna (RMF) collection [25]. BG-Sentinel traps were placed outside houses in the evening at 7 p.m. and collected again in the morning at 6 a.m. The capture of the residual morning fauna of mosquitoes was carried out by the spray method inside the houses between 6 and 9 a.m. Spray capture consists of spreading white sheets on all surfaces and furniture in the bedroom, and the bedroom is sprayed after windows and doors have been closed. A few minutes later, the dead or stunned mosquitoes on the sheets are collected. Each day, the collected mosquitoes were identified morphologically at genus and/or species level using morphological keys [26,27] and then preserved individually in 1.5 mL Eppendorf tubes containing silica gel. At the end of the collection, all mosquitoes were sent to the laboratory at IHU Méditerranée Infection in Marseille, France for further analyses.

### 2.2. Molecular Identification of Mosquitoes

Six mosquito specimens morphologically identified as one *An. gambiae*, two *An. funestus,* and fifteen *Culex* sp. and the mosquitoes whose MALDI-TOF MS spectra that had been selected for inclusion in the MALDI-TOF MS database were submitted for molecular analysis to confirm theit identification to the species level. Similarly, blood from the engorged abdomen of three females *An. gambiae* were submitted for molecular analysis to determine the origins of their blood meals.

Three specimens of *Mansonia uniformis* with good quality spectra and identified with a score greater than or equal to 1.7 when their spectra were tested against the upgraded MS arthropod database (see below), but which presented a discrepancy between the morphological and MALDI-TOF MS identification, were also submitted for molecular analysis. 

DNA was extracted from the cephalothorax of the mosquito and 10 µL of supernatant from crushed engorged female abdomens as described previously [20]. DNA extractions were performed using the EZ1 DNA Tissue Kit (Qiagen, Hilden, Germany) according to the manufacturer’s instructions, as described previously [21,22]. 

For mosquito identification, a 720-bp fragment of the cytochrome C oxidase I gene (mCOI) was amplified by PCR using the following primers: (LCO1490 (forward): 5′-GGTCAAC AAATCATAAGATATTGG-3′; HC02198 (reverse): 5′- TAAACTTCAGGGTGACCAAAAAATCA-3′ [13]. For molecular identification of the blood meal, we performed PCR using the vertebrate cytochrome C oxidase (vCOI) gene that amplifies a 650-bp fragment with the following primers: (vCOI_long (forward): 5′-AAGAATCAGAATARGTGTTG-3′; vCOI_long (reverse): 5′-AACCACAAAGACATTGGCAC-3′) [28]. The sequences obtained were assembled and analysed using ChromasPro software (version 1.7.7) (Technelysium Pty. Ltd., Tewantin, Australia), and then compared to reference sequences available in GenBank (http://blast.ncbi.nlm.nih.gov/, accessed on 25 March 2022). The sequences not identified to the species level by this method were submitted to the Barcode of Life Data Systems: BOLD Systems database (https://www.boldsystems.org/, accessed on 25 March 2022).

### 2.3. Preparation of Mosquitoes for MALDI-TOF MS Analysis

#### 2.3.1. MALDI-TOF MS Identification of Mosquitoes

For MALDI-TOF MS analysis, the legs of each mosquito were placed in a 1.5 mL Eppendorf tube and ground with Tissue Lyser (Qiagen, Hilden, Germany) in a mix solution containing 15 µL of 70% (*v*/*v*) formic acid (Sigma-Aldrich, St. Louis, MO, USA) and 15µl of 50% (*v*/*v*) acetonitrile (Fluka, Buchs, Switzerland) and a pinch of glass beads (Sigma, Lyon, France) for three minutes divided into three 60 s cycles at a frequency of 30 Hz [28]. The samples were then centrifuged for one minute at 10,000 rpm and 1 µL of the supernatant from each homogenate was deposited in four replicates on a MALDI-TOF MS target plate (Bruker Daltonics, Wissembourg, France). Each spot was then coated with 1 µL of a matrix solution of α-cyano-4-hydroxycynnamic acid (CHCA) composed of 50% acetonitrile (*v*/*v*), 2.5% trifluoroacetic acid (*v*/*v*) (Aldrich, Dorset, UK), saturated α-cyano-4-hydroxycynnamic acid (Sigma, Lyon, France) and high-performance liquid chromatography (HPLC)-grade water [29]. After drying for a few minutes at room temperature, the target plate was introduced into the MALDI-TOF Microflex LT apparatus (Bruker Daltonics, Bremen, Germany) for analysis. *Anopheles coluzzii* legs reared in our laboratory were used as a positive control.

#### 2.3.2. MALDI-TOF MS Identification of Blood Meal Sources

The abdomen of each engorged female mosquito was individually ground with a pestle into an Eppendorf tube containing 50 µL of HPLC-grade water. After centrifugation, 10 µL of supernatant was used for MALDI-TOF MS [20]. This was then mixed with 20 µL of 70% (*v*/*v*) formic acid and 20 µL of 50% (*v*/*v*) acetonitrile (Fluka, Buchs, Switzerland) and then centrifuged at 10,000 rpm for 20 s. One microlitre of supernatant was deposited in four replicates per sample on a MALDI-TOF MS target plate and each spot was then coated with 1 µL of HCCA matrix solution consisting of saturated α-cyano-4-hydroxycynnamic acid (Sigma, Lyon, France), 50% acetonitrile (*v*/*v*), 2.5% trifluoroacetic acid (*v*/*v*) (Aldrich, Dorset, UK), and HPLC-grade water. After drying at room temperature for a few minutes, the target plate was introduced into the Microflex LT MALDI-TOF mass spectrometer (Bruker Daltonics, Germany) for analysis.

### 2.4. MALDI-TOF MS Parameters

The spectrum profiles were obtained using a Microflex LT MALDI-TOF mass spectrometer (Bruker Daltonics, Germany), with linear positive ion mode detection at a laser frequency of 50 Hz in a mass range of 2 kDa to 20 kDa. The accelerating voltage was 20 kV with an extraction time of 200 ns. Each spectrum corresponds to ions obtained from 240 laser shots in six regions of a single spot and acquired automatically using AutoXecute Flex Control v.2.4 software (Bruker Daltonics) [21].

### 2.5. Spectral Analysis and Database Creation

All spectra obtained from mosquito legs and blood from engorged abdomens were visualised using flexAnalysis v.3.3 software to assess their quality, reproducibility, and intensity. The quality of the spectra was evaluated according to their reproducibility, a maximum intensity > 3000 au, and an absence of background noise. All spectra that did not meet these three criteria were considered to be poor and were therefore excluded from the analysis.

Our in-house database of MALDI-TOF MS arthropod spectra was updated with spectra from twenty-three mosquito specimens, including four *An. gambiae*, two *An. arabiensis*, one *An. funestus*, one *An.* cf. *rivulorum*, fourteen *Cx. nebulosus,* and one *Cx. cinereus*. The identification of all of these mosquitoes was confirmed by molecular biology. The spectra of three blood specimens from engorged female abdomens identified by molecular biology as human blood were also added to the MALDI-TOF MS database [21]. Before this upgrade, our database already contained the spectra from 2212 arthropods of various species, including 88 species of mosquitoes [22,23,30] including those of fresh and frozen *An. gambiae* and *An. funestus* from Senegal, as well as 290 blood spectra of 22 mammal species and two bird species [20,21].

### 2.6. Blind Tests to Identify Mosquitoes and Blood Meals

MALDI-TOF MS spectra of mosquito legs and the blood meals of engorged females were queried against our home-made updated database using the MALDI-Biotyper v3.0. software (Bruker Daltonics). The level of reliability of the identification of mosquito species and blood meal origin was determined using log score values (LSV) assigned by MALDI-Biotyper v.3.0 software (Bruker, Daltonics) and ranging from 0 to 3. This value evaluates the similarity between a tested spectrum and the reference spectra by comparing the position of the peaks and their intensity.

## 3. Results

### 3.1. Mosquito Collection

A total of 645 mosquitoes were collected using both methods, including 223 (34.57%) collected using the RMF methods and 422 (65.43%) collected with BG-Sentinel traps. According to morphological criteria, six species and two genera were identified. For mosquitoes captured with RMF, 201 were morphologically identified as *An. gambiae* s.l. (90.13%), 12 as *An. funestus* (5.38%), 1 as *An. rufipes* (0.44%), and 9 as *Culex* sp. (4.04%). Mosquitoes captured with the BG-Sentinel traps were identified morphologically as 6 *An. gambiae* (1.42%), 1 *An. rufipes* (0.24%), 1 *Anopheles nili* (0.24%), 3 *Anopheles ziemanni* (0.71%), 24 *M. uniformis* (5.69%), 382 *Culex* sp. (90.52%), and 5 *Aedes* sp. (1.18%) (Table 1). This difference in the composition of the species collected by the two sampling methods would be due partly to the collection methods and partly to the collection location (outside or inside) of the chambers.

### 3.2. Molecular Identification of Mosquitoes and Blood Meals

A total of 26 mosquito specimens, 6 of which were identified morphologically as *An. gambiae*, 2 as *An. funestus*, 3 as *M. uniformis,* and 15 as *Culex* sp., as well as three blood meals from engorged female abdomens, were subjected to standard PCR and sequencing. For the spectra from mosquitoes that were used to update our in-house database, BLAST analysis of the sequences from the six mosquitoes morphologically identified as *An. gambiae* showed that four sequences were 99.23–99.61% identical to the *An. gambiae* sequences available in the GenBank (GenBank: MG753747, MT375222, and MG753701) and two were 99.81% identical to *An. arabiensis* sequence (GenBank: MK628484)). For the two specimens identified morphologically as *An. funestus,* one obtained sequence was 99.61% identical to *An. funestus* (GenBank: MT375219) and one was 98.07% identical to *An*. cf. *rivulorum* (GenBank: MT375227). Of the 15 specimens identified morphologically as *Culex* sp., the sequences obtained from 14 showed between 98.74% and 100% identity with sequences of *Culex nebulosus* (Bold Systems) and one was 100% identical to *Culex cinereus* (Bold Systems) (Table 2). The 14 sequences identified as *Cx. nebulosus* using Bold Systems were deposited in GenBank under accession numbers ranging from OQ257031 to OQ257044.

Molecular identification of the blood meals of three engorged mosquito abdomens, the spectra of which were entered into the database, showed that they had 99.75% and 99.83% identity with human blood reference sequences (GenBank: KM101695) (Table 2).

The sequences obtained from the three specimens identified morphologically as *M. uniformis* were identical to the sequences of *Lutzia tigripes* (GenBank: LC507871), with percentages of between 97.57% and 98.53%.

### 3.3. MALDI-TOF MS Identification of Mosquitoes

A total of 582/645 specimens (90.23%) had at least four legs that were used for MALDI-TOF MS analyses. Visualisation of the spectra of the 582 mosquitoes using flexAnalysis v.3.3 software revealed that 329 specimens (56.52%) were of good quality (maximum intensity > 3000 au, baseline subtraction correct, and absence of background noise) (Figure 2). These 329 mosquito spectra were then subjected to further analysis: these mosquitoes included 83 *An. gambiae*, 10 *An. funestus*, 3 *M. uniformis*, 231 *Culex* sp. and 2 *Aedes* sp., based on our morphological identification (Table 2).

The spectra obtained from the legs of the 306 mosquitoes were queried against the home-made MALDI-TOF MS arthropod spectra database upgraded with the spectra of four *An. gambiae*, two *An. arabiensis*, one *An. funestus*, one *An.* cf. *rivulorum*, fourteen *Cx. nebulosus,* and one *Cx. cinereus,* the identification of which had been confirmed by molecular biology. The spectra used to update the database are available at the following link: https://doi.org/10.35081/s8b9-9510 (accessed on 26 September 2022).

Blind interrogation of this database revealed that all 83 specimens morphologically identified as *An. gambiae* were identified by MALDI-TOF MS as *An. gambiae* (*n* = 64), with LSVs scores ranging from 1.71 to 2.46, *An. coluzzii* (*n* = 12) with scores ranging from 1.75 to 2.19, and *An. arabiensis* (*n* = 1) with a score of 1.8. The eight specimens identified morphologically as *An. funestus* were identified by MALDI-TOF MS as *An. funestus* (*n* = 7) with scores ranging from 1.71 to 1.93 and *An.* cf. *rivulorum* (*n* = 1) with a score of 1.99. Blind testing of the 216 *Culex* sp. were identified by MALDI-TOF MS as *Cx. nebulosus* (*n* = 211), with scores ranging from 1.70 to 2.36, *Cx. quinquefasciatus* (*n* = 2) with scores of 2.21 and 2.31, *Cx. duttoni* (*n* = 1) with a score of 1.77, *Cx. perfescus* with a score of 1.95, and *Cx. tritaeniorhynchus* (*n* = 1) with a score of 1.75. The three specimens identified morphologically as *M. uniformis* were identified by MALDI-TOF MS as *L. tigripes* with scores between 1.74 and 1.87. For *Aedes* sp., two specimens had good quality spectra and were identified as *Aedes aegypti,* with scores of 1.93 and 2 (Table 3).

### 3.4. MALDI-TOF MS Identification of Blood Meals

A total of 157 engorged female abdomens, including *An. gambiae* (*n* = 146), *An. funestus* (*n* = 10) and *An. rufipes* (*n* = 1), were used for MALDI-TOF MS analyses. The MALDI-TOF MS spectra from blood meal analyses revealed that 123 spectra (78.34%) were of good quality (Table 3). After the addition of spectra from three abdomens engorged with human blood (confirmed by molecular biology) to our in-house database, which already contained human as well as mixed-blood spectra [19], the blind test showed that ninety-seven spectra were identified as human blood, six as cow blood, two as dog blood, one as goat blood, one as sheep blood, one as bat blood, two as mixed blood (75% human blood and 25% dog blood), and ten spectra as mixed blood (75% human and 25% sheep blood), with scores ranging from 1.70–2.54 (Table 4).

## 4. Discussion

MALDI-TOF MS has been recognised as having revolutionised clinical microbiology in recent years due to its speed and reliability in the systematic identification of many microorganisms. It has also started to prove its effectiveness at rapidly and accurately identifying arthropods of medical interest, including mosquitoes [9]. Recently, it has been successfully used to identify the origin of blood meals and to detect arthropod-associated microorganisms [21,22,31,32]. Nevertheless, the application of this method requires a protocol readjustment depending on the arthropod body part as well as on the conservation method [29]. Generally, for the optimal preservation of field-collected arthropods and easy transportation to laboratories, storage in 70 °C ethanol or silica gel is more appropriate, due to low cost. Recent studies have shown the effectiveness of MALDI-TOF MS in identifying mosquitoes stored in silica gel for a short time [22].

In this study, 56.52% of good-quality spectra were obtained from the legs of mosquitoes collected and then tested by MALDI-TOF MS. Nevertheless, recent studies on mosquitoes preserved in silica gel generated a higher percentage of good quality spectra compared to our study [22]. We speculate that this could be due to the length of time the mosquitoes were stored in silica gel, as the specimens collected in the recent study [22]. were only stored for two months before being tested, while those in our study were stored for over six months in silica gel. In future studies, we would also consider washing the mosquitoes in ethanol or bleach before storing them in silica gel, in order to remove any traces of fungi, for example, that could possibly damage the mosquito. To minimize the impact of storage time, it would be a good idea to transfer this MALDI-TOF MS database to Senegal, so as to be able to identify mosquitoes on site in future studies.

In this study, all mosquitoes with good-quality spectra were identified by MALDI-TOF MS with scores of between 1.70 and 2.46. This enabled us to identify 12 species of mosquitoes as well as to add four mosquito species spectra to our database. MALDI-TOF MS confirmed the morphological identification of two species, *An. gambiae* and *An. funestus*. It showed its ability to distinguish *An. gambiae*, *An. coluzzii*, and *An. arabiensis* from species of the same complex, which are difficult to differentiate morphologically, as reported in previous studies [33]. It was able to identify species of *Culex* sp., including *Cx. nebulosus*, *Cx. quinquefasciatus*, *Cx. perfescus*, *Cx. duttoni,* and *Cx. tritaeniorhynchus,* as well as species of *Aedes* sp., such as *Ae. aegypti*. Inconsistencies were observed between morphological identification and MALDI-TOF MS identification for some species, such as *An. funestus* and *M*. *uniformis*. Specimens identified morphologically as *An. funestus* were identified by MALDI-TOF MS as *An*. cf. *rivulorum*, a species that was added to our database after confirmation by molecular biology. All specimens identified morphologically as *M. uniformis* were identified by MALDI-TOF MS as *L. tigripes* and confirmed by molecular biology. These errors related to the morphological identification, which was corrected through molecular biology, demonstrating the limits of morphological identification as discussed earlier, but also the reliability and the robustness of MALDI-TOF MS. Despite the difficulty of morphological identification in future studies, it would be a good idea to try to identify all *Culex* sp. specimens at least to species level, to enable us to assess our percentage reliability of identification. Given that the composition of the mosquito fauna in our study is largely inferior to that already reported [34] from the locality, for future studies it would be advisable to use other methods such as CDC light traps, but also to carry out collections over a longer period. 

In this study, molecular biology confirmed the morphological and/or MALDI-TOF MS identification of some specimens, notably those added to the database and those presenting inconsistencies between morphological and MALDI-TOF MS identification. Before adding the spectra of *An. gambiae* and *An. funestus* to our database, the database already contained spectra of fresh or frozen specimens of these two species. We then hypothesised that there may be variations in spectral profiles within the same species, depending on the method of storage. 

All the mosquito species identified in this study have been previously reported in Senegal [7]. Some of them are medically important mosquitoes, as they are involved in several infectious diseases of concern to public health [1]. The species of the gambiae complex (*An. gambiae, An. arabiensis,* and *An. coluzzii*) and the *An. funestus* group (*An. funestus*) identified in this study are recognised as the main vectors of malaria in Senegal [35,36,37]. Among them, *An. gambiae* is predominantly represented in this study. During our study, we were able to identify several species of *Culex* sp., including *Cx. nebulosus*, *Cx. quinquefasciatus*, *Cx. duttoni*, *Cx. perfescus,* and *Cx. tritaeniorhynchus*. Some of them are potential vectors of West Nile fever, Japanese encephalitis, and lymphatic filariasis [2]. In Senegal, *Cx. tritaeniorhynchus* has been shown to be a primary vector of West Nile fever [38]. *Cx. quinquefasciatus* has been recognised by some studies as a potential vector of Rift Valley fever and West Nile fever [39,40]. *Ae. aegypti* is the primary vector of yellow fever, Zika, Dengue, and Chikungunya viruses in humans [41,42,43]. *M. uniformis* has been implicated in the transmission of West Nile and Rift Valley fever [44,45].

In this study, we obtained 78.76% of good-quality spectra from engorged *An. gambiae* abdomens. MALDI-TOF MS analysis of these spectra showed that 82.14% were correctly identified as human blood. A total of 70% of good-quality spectra were obtained from engorged *An. funestus* abdomen spectra. The majority of these spectra (57.14%) were identified by MALDI-TOF MS as human blood. MALDI-TOF MS was able to identify all good-quality abdomen spectra obtained in this study and revealed the majority presence of human blood. This could be explained by the fact that these mosquitoes, including *An. gambiae* and *An. funestus,* are highly anthropophilic and endophilic [46]. *An. rufipes* has been found engorged with human blood and has been reported in the literature to feed on domestic animals, but can accidentally bite humans [26]. During our study, we also identified mixed-blood spectra in *An. gambiae* and *An. funestus*. Of these spectra, 1.78% were identified as mixed blood (75% human blood and 25% dog blood) and 8.92% were identified as mixed blood (75% human blood and 25% sheep blood). These results confirm the data published by the recent study on the ability of MALDI-TOF MS to identify mixed-blood meals [47].

## 5. Conclusions

This study confirmed the reliability, speed, and robustness of the MALDI-TOF MS tool to identify wild mosquitoes preserved in silica gel for a long time. In this study, we applied MALDI-TOF MS for the first time to identify mosquitoes captured in Senegal and the origin of their blood meals. Our research confirms the ability of MALDI-TOF MS to correct errors related to the morphological identification of arthropods. Although all species were correctly and accurately identified, the rate of good-quality spectra obtained in this study appears to be low. It would be interesting to apply this method to mosquitoes that are first cleaned with ethanol or bleach before being preserved in silica gel or to transfer the MALDI-TOF MS database to Senegal, so that the mosquitoes can be processed on site as quickly as possible.

## Figures and Tables

**Figure 1 insects-14-00785-f001:**
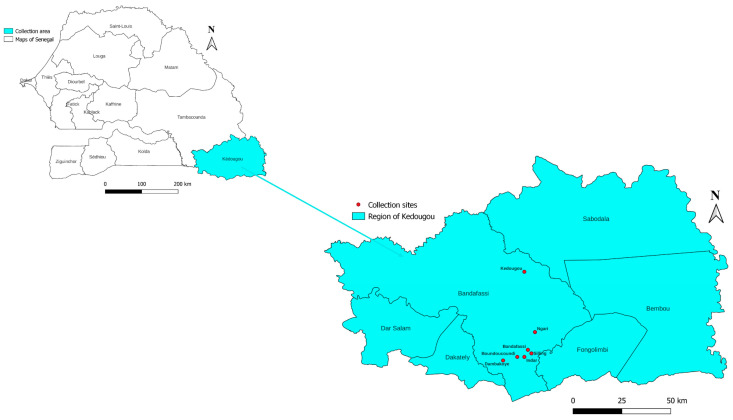
Map of Kédougou (Senegal) showing mosquito collection sites prepared using QGIS version 3.10. The layers have been uploaded to the DIVA-GIS website: https://www.diva-gis.org/datadown, accessed on 25 March 2022.

**Figure 2 insects-14-00785-f002:**
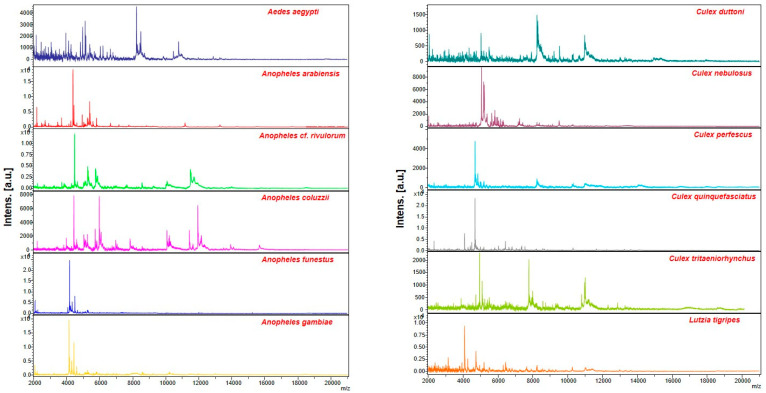
Representative MS profiles obtained for each species of mosquitoes collected in Senegal using flexAnalysis 3.3 software au: arbitrary units, m/z: mass-to-charge ratio.

**Table 1 insects-14-00785-t001:** Number of mosquitoes collected in different villages in Kédougou (Senegal) by two capture methods in 2019.

Method of Collection	RMF						BG—SENTINEL						Total Specimens Collected RMF/BG- Sentinel
Collection Sites	Bandafassi	Boundoucoundi	Indar	Silling	Ngari	Dambakoye	Kédougou	Bandafassi	Dambakoye	Indar	Silling	Ngari	
*Anopheles gambiae*	6	18	17	12	21	127	6						201/6
*Anopheles rufipes*	1									1			1/1
*Anopheles funestus*	1	4		3		3							11/-
*Anopheles nili*												1	-/1
*Anopheles ziemanni*							1				2		-/3
*Mansonia uniformis*							3		9		12		-/24
*Culex* sp.	1			5		3	136	6	33	72	122	13	9/382
*Aedes* sp.							5						-/5
Total	9	22	17	20	21	133	151	6	42	73	136	14	222/422

**Table 2 insects-14-00785-t002:** Results of molecular identification of mosquito specimens from Senegal, and engorged abdomens with the number of spectra added to the database before blind testing the remaining mosquitoes.

Method of Collection	Sample Number	Morphological Identification	Molecular Identification	Percentage Identity (%)	Accession Number
Residual Morning Fauna	157	*An. gambiae*	*An. gambiae*	99.23	MG753747
	55	*An. gambiae*	*An. gambiae*	99.61	MT375222
	31	*An. gambiae*	*An. gambiae*	99.61	MT375222
	60	*An. gambiae*	*An. gambiae*	99.61	MG753701
	98	*An. gambiae*	*An. arabiensis*	99.81	MK628484
	136	*An. gambiae*	*An. arabiensis*	99.81	MK628484
	67	*An. funestus*	*An. funestus*	99.61	MT375219
	257	*An. funestus*	*An.* cf. *rivulorum*	98.07	MT375227
	15 (Abdomen)	Engorged *An. gambiae*	*Homo sapiens*	99.75–99.83	KM101695
	74 (Abdomen)				
	76 (Abdomen)				
BG-Sentinel	45	*Culex* sp.	*Culex nebulosus*	98.74–100	
	163	*Culex* sp.			
	186	*Culex* sp.			
	230	*Culex* sp.			
	235	*Culex* sp.			
	238	*Culex* sp.			
	285	*Culex* sp.			
	286	*Culex* sp.			
	291	*Culex* sp.			
	305	*Culex* sp.			
	318	*Culex* sp.			
	324	*Culex* sp.			
	340	*Culex* sp.			
	346	*Culex* sp.			
	232	*Culex* sp.	*Culex cinereus*	100	

**Table 3 insects-14-00785-t003:** Number of specimens collected and used for MALDI-TOF MS analysis and blind test results for MS identification of each species of mosquitoes collected in Senegal.

Mosquitoes Species Morphologically Identified	Number of Specimens Collected	Number of Specimens with Legs	Number of Spectra of Good Quality	Number of Spectra Added to Database	MALDI-TOF MS Identification	Log Score Value (Average)
*An. gambiae*	207	175/207 (84.54%)	83/175	6	*An. gambiae* (64)	[1.71–2.46] (2.03)
					*An. coluzzii* (12)	[1.75–2.19] (1.89)
					*An. arabiensis* (1)	1.81
*An. rufipes*	2	1/2 (50%)	/	/		
*An. funestus*	12	11/12 (91.66%)	10/11	2	*An.funestus* (7)	[1.71–1.93] (1.78)
					*An.* cf. *rivulorum* (1)	1.99
*An. nili*	1	1/1 (100%)	/	/		
*An. ziemanni*	3	3/3 (100%)	/	/		
*M. uniformis*	24	20/24 (83.33%)	3/20	/	*Lutzia tigripes* (3)	[1.74–1.87] (1.77)
*Culex* sp.	391	368/391 (94.11%)	231/368	15	*Culex nebulosus* (211)	[1.70–2.36] (2.03)
					*Culex quinquefasciatus* (2)	[2.21–2.31] (2.26)
					*Culex duttoni* (1)	1.77
					*Culex perfescus* (1)	1.95
					*Culex tritaeniorhynchus* (1)	1.75
*Aedes* sp.	5	3/5 (60%)	2/3	/	*Aedes aegypti* (2)	[1.93–2] (1.97)
Total	645	582/645 (90.23%)	329/582	23		

**Table 4 insects-14-00785-t004:** Number of engorged mosquitoes from Senegal used for MALDI-TOF MS identification of blood meal origin and blind test results.

Mosquito Species or Genus	Number of Specimens Collected	Number of Spectra of Good Quality	Number of Spectra Added in Database	Identification of Mosquito Blood Meal Sources by MALDI-TOF MS	Log Score Value (Average)
Engorged *An. gambiae*	146	115/146	3	Human (92)	[1.70–2.54] (2.02)
				Cow (4)	[1.70–1.89] (1.75)
				Dog (2)	[1.76–1.87] (1.81)
				Goat (1)	2.04
				Sheep (1)	1.85
				75% Human/25% Dog (2)	[2.10–2.11] (2.10)
				75% Human/25% Sheep (10)	[1.93–2.31] (2.17)
Engorged *An. funestus*	10	7/10	/	Human (4)	[1.85–2.09] (2)
				Cow (2)	[1.86–2.01] (1.93)
				Bat (1)	1.77
Engorged *An. rufipes*	1	1/1	/	Human (1)	2.19
Total	157	123/157	3		

## Data Availability

The sequences of mosquitoes identified as *Cx. nebulosis* are available on GenBank under the numbers OQ257031 to OQ257044. The spectra used to update the database are available at the following link: https://doi.org/10.35081/s8b9-9510 (accessed on 26 September 2022).

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
