# Peer review of "Using MALDI-TOF MS to Identify Mosquitoes from Senegal and the Origin of Their Blood Meals"

_insects, 2023, doi:10.3390/insects14100785_

Round 1
Reviewer 1 Report
This manuscript describes the use of MALTI-TOF MS to identify mosquitoes and their bloodmeals from field-caught mosquitoes in southeastern Senegal.
11. More details on the mosquito collections should be provided. How were the residual morning fauna collections conducted? Were collections made indoors and/or outdoors?
22. There were large differences in species composition between the two sampling methods (RMF and BG Sentinel traps). Can the authors provide some explanation for this?
33. How representative were the collections of the mosquito fauna in the study sites? It is surprising that only 5 Aedes specimens were collected, given that several Aedes species occur in the area, and Aedes aegypti is a prevalent species in these rural and urban study sites.
44. The Discussion is largely a recapitulation of the Results. It would be good to include in the Discussion how these results can or will guide future studies.
55. What was the percentage agreement among the three mosquito identification methods (morphology, molecular and MALDI-TOF)?
Author Response
Manuscript ID: insects-2570800
Type of manuscript: Article
Title: Using MALDI-TOF MS to identify mosquitoes from Senegal and the origin of their blood meals
Authors: Fatou Kiné Fall, Adama Zan Diarra, Charles Bouganali, Cheikh Sokhna, Philippe PAROLA *
*Corresponding author: Philippe Parola
Reviewer 1
Comments and Suggestions for Authors
This manuscript describes the use of MALTI-TOF MS to identify mosquitoes and their bloodmeals from field-caught mosquitoes in southeastern Senegal.
- More details on the mosquito collections should be provided. How were the residual morning fauna collections conducted? Were collections made indoors and/or outdoors?
Answer: Thank you for your comment. Residual morning mosquito fauna was collected using the spray capture method inside bedrooms between 6 am and 9 am and BG-Sentinel traps were placed outside houses in the evening at 7 pm and collected again in the morning at 6 am. We have added the following sentence to the manuscript line 78-83 “BG-Sentinel traps were placed outside houses in the evening at 7pm and collected again in the morning at 6am. The capture of the residual morning fauna of mosquitoes was carried out by the spray method inside the houses between 6 and 9 am. Spray capture consists of spreading white sheets on all surfaces and furniture in the bedroom and the bedroom is sprayed after windows and doors have been closed. A few minutes later, the dead or stunned mosquitoes on the sheets are collected.”
- There were large differences in species composition between the two sampling methods (RMF and BG Sentinel traps). Can the authors provide some explanation for this?
Answer: Thank you for your comment, we believe that this difference in species composition between the two sampling methods is due partly to the collection methods and partly to the location (outdoor and indoor) of the chambers. The first method (BG-Sentinel traps) captures more exophilic mosquitoes, while the second (spray capture) captures endophilic mosquitoes. We have added the following sentence to the manuscript in result section line 193-195 “This difference in the composition of the species collected by the two sampling methods would be due partly to the collection methods and partly to the collection location (outside or inside) of the chambers.”
- How representative were the collections of the mosquito fauna in the study sites?
Answer: We thank the reviewer, our collection represents only 11% of the mosquito fauna in the study sites if we take into account the results of this study (Comparative population dynamics of Culicidae in Kédougou (Sudano-Guinean zone) and Barkédji (Sahelian savannah zone): consequences for arbovirus transmission). However, the techniques and collection period of this study and ours are different.
It is surprising that only 5 Aedes specimens were collected, given that several Aedes species occur in the area, and Aedes aegypti is a prevalent species in these rural and urban study sites.
Answer: You're right, but we thought this was due to the fact that our collection was carried out at times when Aedes mosquitoes weren't too active.
- The Discussion is largely a recapitulation of the Results. It would be good to include in the Discussion how these results can or will guide future studies.
Answer: Thank you, we have added passages in the discussion referring to how our results can or will guide future studies as follows
in line 287-292 “In future studies we would also consider washing the mosquitoes in ethanol or bleach before storing them in silica gel, in order to remove any traces of fungi, for example, that could possibly damage the mosquito. To minimize the impact of storage time, it would be a good idea to transfer this MALDI-TOF MS database to Senegal, so as to be able to identify mosquitoes on site in future studies.”
In line 310-315 “Despite the difficulty of morphological identification in future studies, it would be a good idea to try to identify all Culex sp. specimens at least to species level, to enable us to assess our percentage reliability of identification. Given that the composition of the mosquito fauna in our study is largely inferior to that already reported 34. from the locality, for future studies it would be advisable to use other methods such as CDC light traps, but also to carry out collections over a longer period.”
In line 337-362 “Although all species were correctly and accurately identified, the rate of good quality spectra obtained in this study appears to be low. It would be interesting to apply this method to mosquitoes that are first cleaned with ethanol or bleach before being preserved in silica gel or to transfer the MALDI-TOF MS database to Senegal, so that the mosquitoes can be processed on site as quickly as possible.”
- What was the percentage agreement among the three mosquito identification methods (morphology, molecular and MALDI-TOF)?
Answer: Thank you, but it's difficult to answer this question accurately because not all specimens have been morphologically identified to species level, not all have been subjected to morphological identification and not all have been identified by MALDI-TOF MS because they don't have the right spectrum.
Reviewer 2 Report
Dear Authors,
Submitted Manuscript is very interesting and written in good English. The MS has several technical mistakes with need to be addressed.
Please find my suggestion below.
Abstract
There are full stops after full name of genus, such as Culex. quinquefasciatus (n=2), Culex. duttoni (n=1), Culex. perfescus (n=1), Culex. tritaeniorhynchus (n=1). Please correct it.
L26 Please correct Aegypti to aegypti
L43 It should be [2,5,6]
L77 Please correct Fig1
L77 It is not clear how the authors sampled mosquitoes. Both methods were applied 6-9 AM? How often were methods applied? How did you collect Anopheles during the day when they are active during the night?
L81 Identification key reference (26) is only for Anopheles. What did authors use for other genera?
L83 Please add France.
L181 Gender of genera? If you want to say gender, then you should give how many males and how many females.
L187 Missing space Table1
Table 1 is in a very bad format. The authors should format it so that it is possible to follow the numbers. In this version is not easy to read it.
L204 It is lack of the analyses that authors did not identified Culex to the species level because that would give the % of precision of entomologist who is doing morphological ID. It would give an answer to these experts how reliable are some identifications, not because of their experience but because of the conditions of samples but also some species are very difficult by themselves to be identified.
Table 2. It is not clear the number of engorged females that authors gave in text is three and in the Table 2 there are higher numbers (15, 74, 76). What is the number of analyzed females? If these numbers in bracket are number of blood fed females then is my question why did authors only analyze only 3? Please clarify that.
In that table is also unclear what the number of sample is? Number of mosquitoes? Number of samples?
If that is just a code than it is not needed to give so many rows for the same species. It could be given in one row with the range of percentage from the lowest to the highest.
Table 3. It should be written that first column are mosquitoes species morphologically identified.
English does not require significxan changes.
Author Response
Manuscript ID: insects-2570800
Type of manuscript: Article
Title: Using MALDI-TOF MS to identify mosquitoes from Senegal and the origin of their blood meals
Authors: Fatou Kiné Fall, Adama Zan Diarra, Charles Bouganali, Cheikh Sokhna, Philippe PAROLA *
*Corresponding author: Philippe Parola
Reviewer 2
Comments and Suggestions for Authors
Dear Authors,
Submitted Manuscript is very interesting and written in good English. The MS has several technical mistakes with need to be addressed.
Please find my suggestion below.
Abstract
There are full stops after full name of genus, such as Culex. quinquefasciatus (n=2), Culex. duttoni (n=1), Culex. perfescus (n=1), Culex. tritaeniorhynchus (n=1). Please correct it.
Answer: Thank you for your comment, as suggested we have corrected in line 25-26
L26 Please correct Aegypti to aegypti
Answer: Thank you for your comment, as suggested we have corrected in line 26
L43 It should be [2,5,6]
Answer: Thank you for your comment, we have corrected in line 43
L77 Please correct Fig1
Answer: Thank you for your comment, as suggested we have corrected in line 76
L77 It is not clear how the authors sampled mosquitoes. Both methods were applied 6-9 AM?
Answer: Thank you for your comment, No, both methods were applied from 6 to 9 a.m. Only the collection of residual morning fauna by the bedroom spraying method was carried out between 6 and 9 a.m. BG-Sentinel traps were placed outside houses at 7pm and collected again at 6am. We have clarified this in the manuscript as follows in line 78-83 “BG-Sentinel traps were placed outside houses in the evening at 7 pm and collected again in the morning at 6am. The capture of the residual morning fauna of mosquitoes was carried out by the spray method inside the houses between 6 and 9 am. Spray capture consists of spreading white sheets on all surfaces and furniture in the bedroom and the bedroom is sprayed after windows and doors have been closed. A few minutes later, the dead or stunned mosquitoes on the sheets are collected”
How often were methods applied?
Answer: Both methods were applied daily for one week (7 days) and we have added this in line 72-73 “All mosquitoes were collected in the Kédougou region of Senegal during one week (seven day) in September 2019.”
How did you collect Anopheles during the day when they are active during the night?
Answer: We collected them using the spray method, which allows us to collect Anopheles that are resting inside the bedrooms and not those that are active.
L81 Identification key reference (26) is only for Anopheles. What did authors use for other genera?
Answer: Thank you for your comment. As suggested, we have added to the text the reference on which we based our identification of other genres, as follows:
Line 85: " Each day, the collected mosquitoes were identified morphologically at genus and/or species level using morphological keys [26, 27], and then preserved individually in 1.5 ml Eppendorf tubes containing silica gel."
Line 436-437: "27. Edwards, F.W. Mosquitoes of the Ethiopian Region. III.- Culicine Adults and Pupae. Annals of the Entomological Society of America. 1942, Volume 35, Issue 4, page 475."
L83 Please add France.
Answer: Thank you for your comment. As suggested, we have added France to the text as follows:
Line 87: “At the end of the collection, all mosquitoes were sent to the laboratory at IHU Méditerranée Infection in Marseille, France for further analysis”
L181 Gender of genera? If you want to say gender, then you should give how many males and how many females.
Answer: Thank you for your comment, as suggested we have corrected as follows:
Line 188: “six species and two genera were identified.”
L187 Missing space Table1
Answer: Thank you for your comment, as suggested we have corrected as follows:
Line 187: " (Table 1). "
Table 1 is in a very bad format. The authors should format it so that it is possible to follow the numbers. In this version is not easy to read it.
Answer: Thank you for your comment. As suggested, we have formatted the table 1.
L204 It is lack of the analyses that authors did not identified Culex to the species level because that would give the % of precision of entomologist who is doing morphological ID. It would give an answer to these experts how reliable are some identifications, not because of their experience but because of the conditions of samples but also some species are very difficult by themselves to be identified.
Answer: You're absolutely right, we'll be taking this comment into account in future studies by trying to identify Culex sp. morphologically.
Table 2. It is not clear the number of engorged females that authors gave in text is three and in the Table 2 there are higher numbers (15, 74, 76). What is the number of analyzed females? If these numbers in bracket are number of blood fed females then is my question why did authors only analyze only 3? Please clarify that.
Answer: Thank you for your comment, we have submitted blood from the abdomen of 3 engorged females to molecular biology and the numbers 15, 74 and 76 represent the numbers assigned to these samples. For greater clarity, we have modified Table 2 by replacing “number of samples “ with “sample number”.
In that table is also unclear what the number of sample is? Number of mosquitoes? Number of samples?
Answer: Thank you for your comment, we have clarified this in Table 2. This is the sample number
If that is just a code than it is not needed to give so many rows for the same species. It could be given in one row with the range of percentage from the lowest to the highest.
Answer: Thank you for your comment, as recommended, we have indicated the species name in a single line with the percentage range from lowest to highest in table 2 as follows:
|
Method of collection |
Sample number |
Morphological identification |
Molecular identification |
Percentage identity (%) |
Accession number |
|
Residual Morning Fauna |
157 |
An. gambiae |
An. gambiae |
99.23 |
MG753747 |
|
55 |
An. gambiae |
An. gambiae |
99.61 |
MT375222 |
|
|
31 |
An. gambiae |
An. gambiae |
99.61 |
MT375222 |
|
|
60 |
An. gambiae |
An. gambiae |
99.61 |
MG753701 |
|
|
98 |
An. gambiae |
An. arabiensis |
99.81 |
MK628484 |
|
|
136 |
An. gambiae |
An. arabiensis |
99.81 |
MK628484 |
|
|
67 |
An. funestus |
An. funestus |
99.61 |
MT375219 |
|
|
257 |
An. funestus |
An. cf. rivulorum |
98.07 |
MT375227 |
|
|
15 (Abdomen) |
Engorged An. gambiae |
Homo sapiens |
99.75 - 99.83 |
KM101695 |
|
|
74 (Abdomen) |
|||||
|
76 (Abdomen) |
|||||
|
BG-Sentinel |
45 |
Culex sp Culex sp Culex sp Culex sp Culex sp Culex sp Culex sp Culex sp Culex sp Culex sp Culex sp Culex sp Culex sp Culex sp Culex sp |
Culex nebulosus |
98.74 - 100
|
|
|
163 |
|||||
|
186 |
|||||
|
230 |
|||||
|
235 |
|||||
|
238 |
|||||
|
285 |
|||||
|
286 |
|||||
|
291 |
|||||
|
305 |
|||||
|
318 |
|||||
|
324 |
|||||
|
340 |
|||||
|
346 |
|||||
|
232 |
Culex cinereus |
100 |
|
Table 3. It should be written that first column are mosquitoes species morphologically identified.
Answer: Thank you for your comment, as suggested we have corrected the first column of Table 3.